# Learning Shared Safety Constraints from Multi-task Demonstrations

**Konwoo Kim** [* 1]   **Gokul Swamy** [* 1]   **Zuxin Liu** [1]   **Ding Zhao** [1]   **Sanjiban Choudhury** [2]   **Zhiwei Steven Wu** [1]

## Abstract

Regardless of the particular task we want them to perform in an environment, there are often shared *safety constraints* we want our agents to respect. For example, regardless of whether it is making a sandwich or clearing the table, a kitchen robot should not break a plate. Manually specifying such a constraint can be both time-consuming and error-prone. We show how to learn constraints from expert demonstrations of safe task completion by extending inverse reinforcement learning (IRL) techniques to the space of constraints. Intuitively, we learn constraints that forbid highly rewarding behavior that the expert could have taken but chose not to. Unfortunately, the constraint learning problem is rather ill-posed and typically leads to overly conservative constraints that forbid all behavior that the expert did not take. We counter this by leveraging diverse demonstrations that naturally occur in multi-task settings to learn a tighter set of constraints. We validate our method with simulation experiments on high-dimensional continuous control tasks.

## 1. Introduction

If a friend was in your kitchen and you told them to "make toast" or "clean the dishes," you would probably be rather surprised if they broke some of your plates during this process. The underlying *safety constraint* that forbids these kinds of behavior is both a) implicit and b) agnostic to the particular task they were asked to perform. Now, let's bring a household robot into the equation, operating within your kitchen. How can we ensure that it adheres to these implicit safety constraints, regardless of its assigned tasks?

One approach might be to write down specific constraints (e.g. joint torque limits) and pass them to the decision-making system of the robot. Unfortunately, more complex

constraints like the ones we consider above are both difficult to formalize mathematically and easy for an end-user to forget to specify (as they would be inherently understood by a human helper). This problem is paralleled in the field of reinforcement learning (RL), where defining reward functions that lead to desirable behaviors for the learning agent is a recurring challenge (Hadfield-Menell et al., 2017). For example, it is rather challenging to handcraft the exact function one should be optimized to be a good driver. The standard solution to this sort of "reward design" problem is to instead demonstrate the desired behavior of the agent and then extract a reward function that would incentivize such behavior. Such *inverse reinforcement learning* (IRL) techniques have found application in fields as diverse as robotics (Silver et al., 2010; Ratliff et al., 2009; Kolter et al., 2008; Ng et al., 2006; Zucker et al., 2011), computer vision (Kitani et al., 2012), and human-computer interaction (Ziebart et al., 2008b; 2012). Given the success of IRL techniques and the similarity between reward and constraint design, we propose extending IRL techniques to the space of constraints. We term such techniques *inverse constraint learning*, or ICL for short.

More formally, we consider a setting in which we have access to demonstrations of the optimal safe policy for a task, along with knowledge about the task's reward. This allows us to look at the difference between the safe optimal and optimal policies for a task. Our first key insight is that ***the actions taken by the optimal but not the safe-optimal policy must be forbidden, allowing us to extract a constraint.***

Unfortunately, the ICL problem is still rather ill-posed. Indeed, prior work in ICL will often learn overly conservative constraints that forbid all behavior the expert did not take (Scobee & Sastry, 2019; Vazquez-Chanlatte et al., 2018; McPherson et al., 2021). However, for tasks in a shared environment with different rewards, there are often safety constraints that should be satisfied regardless of the task (e.g. a plate shouldn't be broken regardless of whether you're serving food on it or cleaning up after a meal). Our second crucial insight is that ***we can leverage multi-task data to provide more comprehensive demonstration coverage over the state space, helping our method avoid degenerate solutions.***

More explicitly, the contributions of our work are three-fold.

---

*Equal contribution [1]Carnegie Mellon University [2]Cornell University. Correspondence to: Gokul Swamy <gswamy@cmu.edu>.

*Proceedings of the Interactive Learning with Implicit Human Feedback Workshop at ICML 2023.*, Honolulu, Hawaii, USA. PMLR 202, 2023. Copyright 2023 by the author(s).

**1. We formalize the inverse constraint learning problem.** We frame ICL as a zero-sum game between a policy player and a constraint player. The policy player attempts to maximize reward while satisfying a potential constraint, while the constraint player picks constraints that maximally penalize the learner relative to the expert. Intuitively, such a procedure recovers constraints that forbid high-reward behavior the expert did not take.

**2. We develop a multi-task extension of inverse constraint learning.** We derive a zero-sum game between a set of policy players, each attempting to maximize a task-specific reward, and a constraint player that chooses a constraint that all policy players must satisfy. Because the constraint player looks at aggregate learner and expert data, it is less likely to select a degenerate solution.

**3. We demonstrate the efficacy of our approach on various continuous control tasks.** We show that with restricted function classes, we are able to recover ground-truth constraints on certain tasks. Even when using less interpretable function classes like deep networks, we can still ensure a match with expert safety and task performance. In the multi-task setting, we are able to identify constraints that a single-task learner would find struggling to learn.

## 2. Related Work

Our work exists at the confluence of various research thrusts. We discuss each independently.

**Inverse Reinforcement Learning.** IRL (Ziebart et al., 2008a;b; 2012; Ho & Ermon, 2016) can be framed as a two-player zero-sum game between a policy player and a reward player (Swamy et al., 2021). In most formulations of IRL, a potential reward function is chosen in an outer loop, and the policy player optimizes it via RL in an inner loop. Similar to IRL, the constraint in our formulation of ICL is chosen adversarially in an outer loop. However, in contrast to IRL, the inner loop of ICL is *constrained* reinforcement learning: the policy player tries to find the optimal policy that respects the constraint chosen in the outer loop.

**Constrained Reinforcement Learning.** Our approach involves repeated calls to a constrained reinforcement learning (CRL) oracle (Garcıa & Fernández, 2015; Gu et al., 2022). CRL aims to find a reward-maximizing policy over a constrained set, often formulated as a constrained policy optimization problem (Altman, 1999; Xu et al., 2022). Solving this problem via Frank-Wolfe methods is often unstable (Ray et al., 2019; Liang et al., 2018). Various methods have been proposed to mitigate this instability, including variational techniques (Liu et al., 2022), imposing trust-region regularization (Achiam et al., 2017; Yang et al., 2020; Kim & Oh, 2022), optimistic game-solving algorithms (Moskovitz et al., 2023), and PID controller-based methods (Stooke

et al., 2020). In our practical implementations, we use PID-based methods for their relative simplicity.

**Multi-task Inverse Reinforcement Learning.** Prior work in IRL has considered incorporating multi-task data (Xu et al., 2019; Yu et al., 2019; Gleave & Habryka, 2018). We instead consider a setting in which we know task-specific rewards and are attempting to recover a shared component of the demonstrator's objective.

**Inverse Constraint Learning.** We are far from the first to consider the ICL problem. Scobee & Sastry (2019); McPherson et al. (2021) extend the MaxEnt IRL algorithm of Ziebart et al. (2008a) to the ICL setting. We instead build upon the moment-matching framework of Swamy et al. (2021), allowing our theory to handle general reward functions instead of the linear reward functions MaxEnt IRL assumes. We are also able to provide performance and constraint satisfaction guarantees on the learned policy, unlike the aforementioned work. Furthermore, we consider the multi-task setting, addressing a key shortcoming of the prior work. In an excellent paper, Chou et al. (2020) also consider the multi-task setting but propose a solution that requires several special solvers. In contrast, our approach is relatively simple to implement on top of an existing IRL implementation and is, therefore, more likely to scale to realistic problems.

In concurrent work, Lindner et al. (2023) propose an elegant solution approach to ICL: rather than learning a constraint function, assume that *any* unseen behavior is unsafe and enforce constraints on the learner to play a convex combination of the demonstrated safe trajectories. The key benefit of this approach is that it doesn't require knowing the reward function the expert was optimizing. However, by forcing the learner to simply replay previous expert behavior, the learner cannot meaningfully generalize, and might therefore be extremely suboptimal on any new task. In contrast, we use the side information of a reasonable set of constraints to provide rigorous policy performance guarantees. Experimentally, their method has only been shown to scale to tabular / linear problems, while our method scales easily to continuous control with deep networks. We also note that, because we scale the learned constraint differently for each task, their impossibility result (Prop. 2) does not apply to our method, thereby elucidating why a naive application of inverse RL isn't sufficient for the problem we consider.

## 3. Formalizing Inverse Constraint Learning

We build up to our full method in several steps. We first describe the foundational algorithmic structures we build upon (inverse reinforcement learning and constrained reinforcement learning). We then describe the single-task formulation before generalizing it to the multi-task setup.

**Algorithm 1** CRL (Constrained Reinforcement Learning)

**Input:** Reward $r$, constraint $c$, learning rates $\eta_{1:N}$, tolerance $\delta$
**Output:** Trained policy $\pi$
Initialize $\lambda_1 = 0$
**for** $i$ in $1 \ldots N$ **do**
  $\pi_i \leftarrow \text{RL}(r = r - \lambda_i c)$
  $\lambda_i \leftarrow [\lambda_i + \eta_i(J(\pi_i, c) - \delta)]^+$
**end for**
**Return** $\text{Unif}(\pi_{1:N})$.

**Algorithm 2** ICL (Inverse Constraint Learning)

**Input:** Reward $r$, constraint class $\mathcal{F}_c$, trajectories from $\pi_E$
**Output:** Learned constraint $c$
Initialize $c_1 \in \mathcal{F}_c$
**for** $i$ in $1 \ldots N$ **do**
  $\pi_i, \lambda_i \leftarrow \text{CRL}(r, c_i, \delta = J(\pi_E, c_i))$
  `# use any no-regret algo. to pick c`
  $c_{i+1} \leftarrow \text{argmax}_{c \in \mathcal{F}_c} \frac{1}{T} \sum_j^i (J(\pi_j, c) - J(\pi_E, c)) + R(c)$.
**end for**
**Return** best of $c_{1:N}$ on validation data.

We consider a finite-horizon Markov Decision Process (MDP) (Puterman, 2014) parameterized by $\langle \mathcal{S}, \mathcal{A}, \mathcal{T}, r, T \rangle$ where $\mathcal{S}, \mathcal{A}$ are the state and action spaces, $\mathcal{T} : \mathcal{S} \times \mathcal{A} \to \Delta(\mathcal{S})$ is the transition operator, $r : \mathcal{S} \times \mathcal{A} \to [-1, 1]$ is the reward function, and $T$ is the horizon.

### 3.1. Inverse RL as Game Solving

In the inverse RL setup, we are given access trajectories generated by an expert policy $\pi^E : \mathcal{S} \to \Delta(\mathcal{A})$, but do not know the reward function of the MDP. Our goal is to nevertheless learn a policy that performs as well as the expert's, no matter the true reward function.

We solve the IRL problem via equilibrium computation between a policy player and an adversary that tries to pick out differences between expert and learner policies under potential reward functions (Swamy et al., 2021). More formally, we optimize over polices $\pi : \mathcal{S} \to \Delta(\mathcal{A}) \in \Pi$ and reward functions $f : \mathcal{S} \times \mathcal{A} \to [-1, 1] \in \mathcal{F}_r$. For simplicity, we assume that our strategy spaces ($\Pi$ and $\mathcal{F}_r$) are convex and compact and that $r \in \mathcal{F}_r, \pi_E \in \Pi$. We solve (i.e. compute an approximate Nash equilibrium) of the two-player zero-sum game

$$\min_{\pi \in \Pi} \max_{f \in \mathcal{F}_r} J(\pi_E, f) - J(\pi, f), \quad (1)$$

where $J(\pi, f) = \mathbb{E}_{\xi \sim \pi}[\sum_{t=0}^{T} f(s_t, a_t)]$ denotes the value of policy $\pi$ under reward function $f$.

### 3.2. Constrained Reinforcement Learning as Game Solving

In CRL, we are given access to both the reward function and a constraint $c : \mathcal{S} \times \mathcal{A} \to [-1, 1]$. Our goal is to learn the highest reward policy that, over the horizon, has a low expected value under the constraint. More formally, we seek a solution to the optimization problem:

$$\min_{\pi \in \Pi} -J(\pi, r) \text{ s.t. } J(\pi, c) \leq \delta, \quad (2)$$

where $\delta$ is some error tolerance. We can also formulate CRL as a game via forming the Lagrangian of the above

optimization problem (Altman, 1999):

$$\min_{\pi \in \Pi} \max_{\lambda > 0} -J(\pi, r) + \lambda(J(\pi, c) - \delta). \quad (3)$$

Intuitively, the adversary updates the weight of the constraint term in the policy player's reward function based on how in violation the learner is.

### 3.3. Single-Task Inverse Constraint Learning

We are finally ready to formalize ICL. In ICL, we are given access to the reward function, trajectories from the solution to a CRL problem, and a class of potential constraints $\mathcal{F}_c$ in which we assume the ground-truth constraint $c^*$ lies. We assume that $\mathcal{F}_c$ is convex and compact.

In the IRL setup, without strong assumptions on the dynamics of the underlying MDP and expert, it is impossible to guarantee recovery of the ground-truth reward. Often, the only reward function that actually makes the expert optimal is zero everywhere (Abbeel & Ng, 2004). Instead, we attempt to find the reward function that maximally distinguishes the expert from an arbitrary other policy in our policy class via game-solving (Ziebart et al., 2008a; Ho & Ermon, 2016; Swamy et al., 2021). Similarly, for ICL, exact constraint recovery can be challenging. For example, if two constraints differ only on states the expert never visits, it is not clear how to break ties. We instead try to find a constraint that best separates the safe and optimal $\pi_E$ from policies that achieve higher rewards.

More formally, we seek to solve the following constrained optimization problem.

$$\min_{\pi \in \Pi} J(\pi_E, r) - J(\pi, r) \quad (4)$$

$$\text{s.t.} \max_{c \in \mathcal{F}_c} J(\pi, c) - J(\pi_E, c) \leq 0. \quad (5)$$

Note that in contrast to the *moment-matching* problem we solve in imitation learning (Swamy et al., 2021), we instead want to be *at least* as safe as the expert. This means that rather than having equality constraints, we have inequality

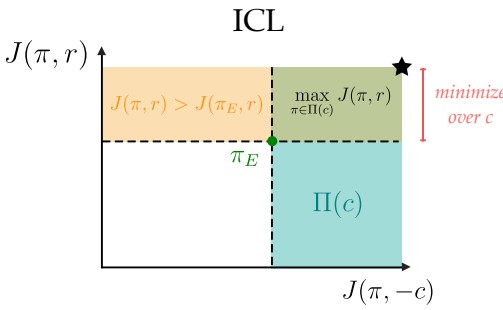

ICL

*Figure 1.* A visual depiction of the optimization problem we're trying to solve in `ICL`. We attempt to pick a constraint that minimizes the value difference over the expert policy a safe policy could have. The star corresponds to the output of CRL.

constraints. Continuing, we can form the Lagrangian:

$$\min_{\pi \in \Pi} \max_{\lambda > 0} J(\pi_E, r) - J(\pi, r) + \lambda(\max_{c \in \mathcal{F}_c} J(\pi, c) - J(\pi_E, c)) \tag{6}$$

$$= \max_{c \in \mathcal{F}_c} \max_{\lambda > 0} \min_{\pi \in \Pi} J(\pi_E, r - \lambda c) - J(\pi, r - \lambda c). \tag{7}$$

Notice that the form of the ICL game resembles a combination of the IRL and CRL games. We describe the full game-solving procedure in Algorithm 2, where $R(c)$ is an arbitrary strongly convex regularizer (McMahan, 2011). Effectively, we pick a constraint function in the same way we pick a reward function in IRL but run a CRL inner loop instead of an RL step. Instead of a fixed constraint threshold, we set tolerance $\delta$ to the expert's constraint violation. Define

$$\ell_i(c) = \frac{1}{T}(J(\pi_i, c) - J(\pi_E, c)) \in [-1, 1] \tag{8}$$

as the per-round loss that the constraint player suffers in their online decision problem. The best-in-hindsight comparator constraint is defined as

$$\hat{c} = \underset{c \in \mathcal{F}_c}{\operatorname{argmax}} \sum_i^T \ell_i(c). \tag{9}$$

We can then define the cumulative regret the learner suffers as

$$\operatorname{Reg}(T) = \sum_i^T \ell_i(\hat{c}) - \sum_i^T \ell_i(c_i), \tag{10}$$

and let $\epsilon_i = \ell_i(\hat{c}) - \ell_i(c_i)$. We prove the following theorem via standard machinery.

**Theorem 3.1.** *Let $c_{1:N}$ be the iterates produced by Algorithm 2 and let $\bar{\epsilon} = \frac{1}{N}\sum_i^N \epsilon_i$ denote their time-averaged regret. Then, there exists a $c \in c_{1:N}$ such that $\pi = CRL(r, c, \delta = J(\pi_E, c))$ satisfies*

$$J(\pi, c^*) - J(\pi_E, c^*) \le \bar{\epsilon}T \text{ and } J(\pi, r) \ge J(\pi_E, r). \tag{11}$$

In words, by optimizing under the recovered constraint, we can learn a policy that (weakly) Pareto-dominates the expert policy under $c^*$. We conclude by noting that because FTRL is a no-regret algorithm for linear losses like (8), we have that $\lim_{T \to \infty} \frac{\operatorname{Reg}(T)}{T} = 0$. This means that with enough iterations, the RHS of the above bound on ground-truth constraint violation will go to 0.

### 3.4. Multi-task Inverse Constraint Learning

---

**Algorithm 3** `MT-ICL` (Multi-task Inverse Constraint Learning)

---

**Input:** Rewards $r^{1:K}$, constraint class $\mathcal{F}_c$, trajectories from $\pi_E^{1:K}$
**Output:** Learned constraint $c$
Set $\widetilde{\mathcal{F}}_c = \{c \in \mathcal{F}_c | \forall k \in [K], J(\pi_E^k, c) \le 0\}$
Initialize $c_1 \in \widetilde{\mathcal{F}}_c$
**for** $i$ in $1 \dots N$ **do**
  **for** $k$ in $1 \dots K$ **do**
    $\pi_i^k, \lambda_i^k \leftarrow \texttt{CRL}(r^k, c_i^k, \delta = 0)$
  **end for**
  `# use any no-regret algo. to pick c`
  $c_{i+1} \leftarrow \operatorname{argmax}_{c \in \widetilde{\mathcal{F}}_c} \frac{1}{TK} \sum_j^i \sum_k^K (J(\pi_j^k, c) - J(\pi_E^k, c)) + R(c).$
**end for**
**Return** best of $c_{1:N}$ on validation data.

---

One of the potential failure modes of the single-task approach we outline above is that we could learn an overly conservative constraint, leading to poor task performance (Liu et al., 2023). For example, imagine that we entropy-regularize our policy optimization (Ziebart et al., 2008a; Haarnoja et al., 2018), as is common practice. Assuming a full policy class, the learner puts nonzero probability mass on all reachable states in the MDP. The constraint player is therefore incentivized to forbid all states the expert did not visit (Scobee & Sastry, 2019; McPherson et al., 2021). Such a constraint would likely generalize poorly when combined with a new reward function ($\tilde{r} \ne r$) as it forbids *all untaken* rather than just *unsafe* behavior.

At heart, the issue with the single-task formulation lies in the potential for insufficient coverage of the state space within expert demonstrations. Therefore, it is natural to explore a multi-task extension to counteract this limitation. Let each task be defined by a unique reward. We assume the dynamics and safety constraints are consistent across tasks. We observe $K$ samples of the form $(r_k, \{\xi \sim \texttt{CRL}(r_k, c^*)\})$. This data allows us to define the multi-task variant of our previously described ICL game:

$$\max_{c \in \mathcal{F}_c} \min_{\pi^{1:K} \in \Pi} \max_{\lambda^{1:K} > 0} \sum_i^K J(\pi_E^i, r^i - \lambda^i c) - J(\pi^i, r^i - \lambda^i c). \tag{12}$$

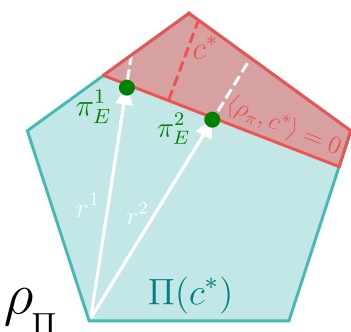

$\rho_\Pi$

*Figure 2.* If we have a sufficient diversity of expert policies, none of which are optimal along the reward vector, we can identify the hyperplane that separates the safe policies from the unsafe policies. The constraint (red, dashed) will be orthogonal to this hyperplane. For this example, because $\rho_\pi \in \mathbb{R}^2$, we need two expert policies.

We describe how we solve this game in Algorithm 3, where $R(c)$ is an arbitrary strongly convex regularizer (McMahan, 2011). In short, we alternate between solving $K$ CRL problems and updating the constraint based on the data from all policies.

We now give two conditions under which generalization to new reward functions is possible.

### 3.5. A (Strong) Geometric Condition for Identifiability

Consider for a moment the linear programming (LP) formulation of reinforcement learning. We search over the space of occupancy measures ($\rho_\pi \in \Delta(\mathcal{S} \times \mathcal{A})$) that satisfy the set of Bellman flow constraints (Sutton & Barto, 2018) and try to maximize the inner product with reward vector $r \in \mathbb{R}^{|\mathcal{S}||\mathcal{A}|}$. We can write the CRL optimization problem (assuming $\delta = 0$ for simplicity) as an LP as well. Using $\rho_\Pi$ to denote the occupancy measures of all $\pi \in \Pi$,

$$\max_{\rho_\pi \in \rho_\Pi} \langle \rho_\pi, r \rangle \text{ s.t. } \langle \rho_\pi, c^* \rangle \leq 0.$$

We observe the solution to such a problem for $K$ rewards, begging the question of when that is enough to uniquely identify $c^*$. Recall that to uniquely determine the equation of a hyperplane in $\mathbb{R}^d$, we need $d$ linearly independent points. $c^* \in \mathbb{R}^{|\mathcal{S}||\mathcal{A}|}$, so we need $|\mathcal{S}||\mathcal{A}|$ expert policies. Furthermore, we need each of these points to lie on the constraint line and not on the boundary of the full polytope. Put differently, we need each distinct expert policy to *saturate* the underlying constraint (i.e. $\exists \pi \in \Pi$ s.t. $J(\pi_E^k, r^k) < J(\pi^k, r^k)$). Under these conditions, we can uniquely determine the hyperplane that separates safe from unsafe policies, to which the constraint vector is orthogonal. More formally,

**Lemma 3.2.** *Let $\pi_E^{1:|\mathcal{S}||\mathcal{A}|}$ be distinct expert policies such that a) $\forall i \in [|\mathcal{S}||\mathcal{A}|]$, $\pi_E^i \in relint(\rho_\Pi)$ and b) no $\rho_{\pi_E^i}$ can be*

*generated by a mixture of the other visitation distributions. Then, $c^*$ is the unique (up to scaling) nonzero vector in*

$$Nul\left(\begin{bmatrix} \rho_{\pi_E^1} - \rho_{\pi_E^2} \\ \vdots \\ \rho_{\pi_E^{|\mathcal{S}||\mathcal{A}|-1}} - \rho_{\pi_E^{|\mathcal{S}||\mathcal{A}|}} \end{bmatrix}\right). \tag{13}$$

We visualize this process for the $|\mathcal{S}||\mathcal{A}| = 2$ case in Fig. 2. Assuming we are able to recover $c^*$, we can guarantee that our learners will be able to act safely, regardless of the task they are asked to do. However, the assumptions required to do so are quite strong: we are effectively asking for our expert policies to form a basis for the space of occupancy measures, which means we must see expert data for a large set of diverse tasks.

Identifiability is too strong a goal as it requires us to estimate the value of the constraint *everywhere* in the state-action space. If we know the learner will only be incentivized to go to a certain subset of states (as is often true in practice), we can guarantee safety without fully identifying $c^*$. Therefore, we now consider how, by making distributional assumptions on how tasks are generated, we can generalize to novel tasks.

### 3.6. A Statistical Condition for Generalization

Assume that tasks $\tau$ are drawn i.i.d. from some $P(\tau)$. Then, even if we do not see a wide enough diversity of expert policies to guarantee identifiability of the ground-truth constraint function, with enough samples, we can ensure we do well in expectation over tasks. For some constraint $c$, let us define

$$V(c) = \mathbb{E}_{\tau \sim P(\tau)}[J(\pi^\tau, c) - J(\pi_E^\tau, c)], \tag{14}$$

where $\lambda^\tau, \pi^\tau = \texttt{CRL}(r^\tau, c)$ denote the solutions to the inner optimization problem. We begin by proving the following lemma.

**Lemma 3.3.** *With*

$$K \geq O\left(\log\left(\frac{|\mathcal{F}_c|}{\delta}\right)\frac{(2T)^2}{\epsilon^2}\right) \tag{15}$$

*samples, we have that with probability $\geq 1 - \delta$, we will be able to estimate all $|\mathcal{F}_c|$ population estimates of $V(c)$ within $\epsilon$ absolute error.*

Note that we perform the above analysis for finite classes but one could easily extend it (Sriperumbudur et al., 2009). The takeaway from the above lemma is that if we observe a sufficient number of tasks, we can guarantee that we can estimate the population loss of all constraints, up to some tolerance.

Consider the learner being faced with a new task they have never seen before at test time. Unlike in the single task case,

where it is clear how to set the cost limit passed to CRL, it is not clear how to do so for a novel task. Hence, we make the following assumption.

**Assumption 3.4.** We assume that $\mathbb{E}_\tau[J(\pi_E^\tau, c^*)] \leq 0$, and that $\forall c \in \mathcal{F}_c, \exists \pi \in \Pi$ s.t. $J(\pi, c) \leq 0$.

This (weak) assumption allows us to a) use a cost limit of 0 for our CRL step and b) search over a subset of $\mathcal{F}_c$ that the expert is safe under. Under this assumption, we are able to prove the following:

**Theorem 3.5.** *Let $c_{1:N}$ be the iterates produced by Algorithm 3 with $K(\epsilon, \delta)$ chosen as in Lemma 3.3 and let $\bar{\epsilon} = \frac{1}{N}\sum_i^N \epsilon_i$ denote their time-averaged regret. Then, w.p. $\geq 1 - \delta$, there exists a $c \in c_{1:N}$ such that $\pi(r) = CRL(r, c, \delta = 0)$ satisfies*

$$\mathbb{E}_{\tau \sim P(\tau)}[J(\pi(r^\tau), c^*) - J(\pi_E^\tau, c^*)] \leq \bar{\epsilon}T + 3\epsilon T \text{ and}$$

$$\mathbb{E}_{\tau \sim P(\tau)}[J(\pi(r^\tau), r^\tau) - J(\pi_E^\tau, r^\tau)] \geq -2\epsilon T.$$

In short, if we observe enough tasks, we are able to learn a constraint that, when optimized under, leads to policies that approximately Pareto-dominate that of the expert.

We now turn our attention to the practical implementation of these algorithms.

# 4. Practical Algorithm

We provide practical implementations of constrained reinforcement learning and inverse constraint learning and benchmark their performance on several continuous control tasks. We first describe the environments we test our algorithms on. Then, we provide results showing that our algorithms learn policies that match expert performance and constraint violation. While it is hard to guarantee constraint recovery in theory, we show that we can recover the ground-truth constraint empirically if we search over a restricted enough function class.

## 4.1. Tasks

We focus on the ant environment from the PyBullet (Coumans & Bai, 2016) and MuJoCo (Todorov et al., 2012) benchmarks. The default reward function incentivizes progress along the positive $x$ direction. For our single-task experiments, we consider a velocity and position constraint on top of this reward function.

1. **Velocity Constraint:** $\frac{\|q_{t+1} - q_t\|_2}{dt} \leq 0.75$ where $q_t$ is the ant's position

2. **Position Constraint:** $0.5x_t - y_t \leq 0$ where $x_t, y_t$ are the ant's coordinates

For our multi-task experiments, we build upon the D4RL (Fu et al., 2020) antmaze benchmark. The default reward function incentivizes the agent to navigate a fixed maze to a random goal position: $\exp(-\|q_{\text{goal}} - q_t\|_2)$. We modify this environment such that the walls of the maze are transparent, but the agent incurs a unit step-wise cost for passing through the maze walls.

Our expert policies are generated by running CRL with the ground-truth constraint. We use the Tianshou (Weng et al., 2022) implementation of PPO (Schulman et al., 2017) as our baseline policy optimizer. Classical Lagrangian methods exactly follow the gradient update shown in Algorithm 1, but they are susceptible to oscillating learning dynamics and constraint-violating behavior during training. The PID Lagrangian method (Stooke et al., 2020) extends the naive gradient update of $\lambda_i$ with a proportional and derivative term to dampen oscillations and prevent cost overshooting. To reduce the amount of interaction required to solve the inner optimization problem, we warm-start our policy in each iteration by behavior cloning against the given expert demonstrations. We used a single NVIDIA 3090 GPU for all experiments. Due to space constraints, we defer all other implementation details to Appendix B.

## 4.2. ICL Results

We begin with results for the single-task problem continuing on to the multi-task setup.

## 4.3. Single-Task Continuous Control Results

As argued above, we expect a proper ICL implementation to learn policies that perform as well and are as safe as the expert. However, by restricting the class of constraints we consider, we can also investigate whether recovery of the ground-truth constraint is possible. To this end, we consider a reduced-state version of our algorithm where the learned constraint takes a subset of the agent state as input. For the velocity constraint, the learned constraint is a linear function of the velocity, while for the position and maze constraints, the learned constraint is a linear function of the ant's position.

Using this constraint representation allows us to visualize the learned constraint over the course of ICL training, as shown in Figure 3. We find that our ICL implementation is able to recover the constraint, as the learned constraint for both the velocity and position tasks converges to the ground-truth value. Our results further show that over the course of ICL training, the learned policies match expert performance as their violations of the ground-truth constraint converge towards the expert's. Figure 4 provides a direct depiction of the evolution of the learned constraint and policy. The convergence of the red and blue lines shows that the learned position constraint approaches the ground truth, and the

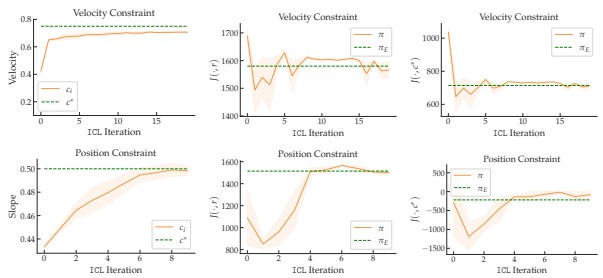

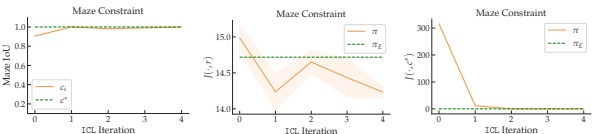

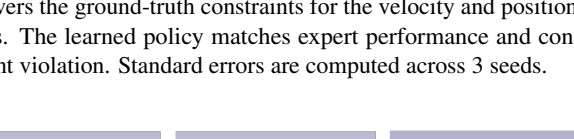

Figure 3. Over the course of training, the learned ICL constraint recovers the ground-truth constraints for the velocity and position tasks. The learned policy matches expert performance and constraint violation. Standard errors are computed across 3 seeds.

Figure 5. We see that over ICL iterations, we are able to recover the ground-truth walls of the ant-maze, enabling the learner to match expert performance and constraint violations. Results for the second two plots are averaged across all 10 tasks. Standard errors are computed across 3 seeds.

Figure 4. As ICL training progresses, the learned position constraint (red line) converges to the ground-truth constraint (blue line) and the policy learns to escape unsafe regions.

policy's behavior approaches that of the expert in response to this.

### 4.4. Multi-Task Continuous Control Results

We next consider an environment where, even with an appropriate constraint class, recovering the ground-truth constraint with a single task isn't feasible due to the ill-posedness of the inverse constraint learning problem. Specifically, we use the AntMaze environment from D4RL (Fu et al., 2020), modified to have a more complex maze structure. As seen in Figure 6, each task is to navigate through the maze from one of the starting positions (top/bottom left) to one of the grid cells in the rightmost column. We provide expert data for all 10 tasks to the learner.

As we can see in Figure 5, multi-task ICL is able to recover the ground-truth maze within a single iteration and quickly learn policies that match expert performance and constraint violation, *all without ever interacting with the ground-truth maze*.

We visually compare several alternative strategies for using the multi-task demonstration data in the bottom row of Figure 6. The 0/1 values in the cells correspond to querying the deep constraint network from the last iteration of ICL on points from each of the grid cells and thresholding at some confidence. We see that a single-task network *(d)* learns spurious walls that would prevent the learner from completing more than half of the tasks. Furthermore, learning

10 separate classifiers and then aggregating their outputs *(e) / (f)* also fails to produce reasonable outputs. However, when we use data from all 10 tasks to train our multi-task constraint network *(g) / (h)*, we are able to perfectly recover the walls of the maze. These results echo our preceding theoretical argument about the importance of multi-task data for learning constraints that generalize to future tasks.

## 5. Discussion

In this work, we derive an algorithm for learning safety constraints from multi-task demonstrations. We show that by replacing the inner loop of inverse reinforcement learning with a constrained policy optimization subroutine, we can learn constraints that guarantee learner safety on a single task. We then give statistical and geometric conditions under which we can guarantee safety on unseen tasks by planning under a learned constraint. We validate our approach on several high-dimensional continuous control tasks.

**Limitations.** On the practical side, our experiments are performed purely on simulated tasks – we would be interested in applying our approach to real-world problems in the future. On the theoretical side, the CRL inner loop can be more expensive than an RL loop – we would be interested in speeding up CRL using expert demonstrations, perhaps by adopting the approach of Swamy et al. (2023). We also ignore all finite-sample issues, which could potentially be addressed via data-augmentation approaches like that of Swamy et al. (2022).

## 6. Acknowledgements

We thank Drew Bagnell for edifying conversations on the relationship between ICL and IRL. ZSW is supported in part by the NSF FAI Award #1939606, a Google Faculty Research Award, a J.P. Morgan Faculty Award, a Facebook Research Award, an Okawa Foundation Research Grant, and a Mozilla Research Grant. KK and GS are supported by a GPU award from NVIDIA.

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

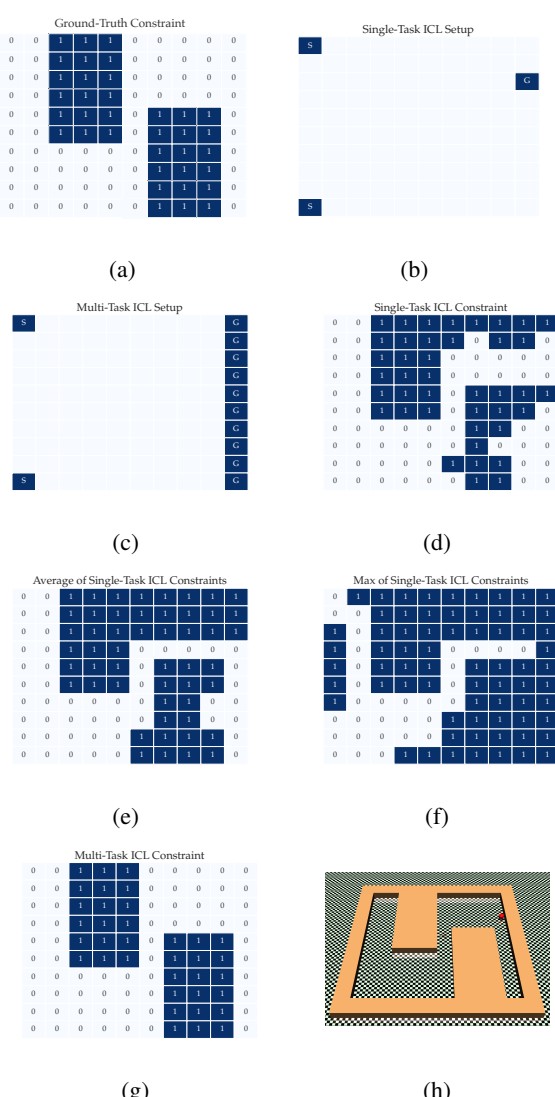

Figure 6. We consider the problem of trying to learn the walls of a custom maze **(a)** based on the AntMaze environment from D4RL (Fu et al., 2020). We consider both a single-task **(b)** and multi-task **(c)** setup. We see that the single-task data is insufficient to learn an accurate constraint **(d)**. Averaging or taking the max over the constraints learned from the data for each of the ten goals **(e)-(f)** also doesn't work. However, if we use the data from all 10 tasks to learn the constraint **(g)-(h)**, we are able to perfectly recover the ground-truth constraint with a single ICL iteration.

Kolter, J. Z., Rodgers, M. P., and Ng, A. Y. A control architecture for quadruped locomotion over rough terrain. In *2008 IEEE International Conference on Robotics and Automation*, pp. 811–818. IEEE, 2008.

Liang, Q., Que, F., and Modiano, E. Accelerated primal-dual policy optimization for safe reinforcement learning. *arXiv preprint arXiv:1802.06480*, 2018.

Lindner, D., Chen, X., Tschiatschek, S., Hofmann, K., and Krause, A. Learning safety constraints from demonstrations with unknown rewards. *arXiv preprint arXiv:2305.16147*, 2023.

Liu, Z., Cen, Z., Isenbaev, V., Liu, W., Wu, S., Li, B., and Zhao, D. Constrained variational policy optimization for safe reinforcement learning. In *International Conference on Machine Learning*, pp. 13644–13668. PMLR, 2022.

Liu, Z., Guo, Z., Cen, Z., Zhang, H., Tan, J., Li, B., and Zhao, D. On the robustness of safe reinforcement learning under observational perturbations. In *International Conference on Learning Representations*, 2023.

McMahan, B. Follow-the-regularized-leader and mirror descent: Equivalence theorems and l1 regularization. In *Proceedings of the Fourteenth International Conference on Artificial Intelligence and Statistics*, pp. 525–533. JMLR Workshop and Conference Proceedings, 2011.

McPherson, D. L., Stocking, K. C., and Sastry, S. S. Maximum likelihood constraint inference from stochastic demonstrations. In *2021 IEEE Conference on Control Technology and Applications (CCTA)*, pp. 1208–1213. IEEE, 2021.

Moskovitz, T., O'Donoghue, B., Veeriah, V., Flennerhag, S., Singh, S., and Zahavy, T. Reload: Reinforcement learning with optimistic ascent-descent for last-iterate convergence in constrained mdps. *arXiv preprint arXiv:2302.01275*, 2023.

Ng, A. Y., Coates, A., Diel, M., Ganapathi, V., Schulte, J., Tse, B., Berger, E., and Liang, E. Autonomous inverted helicopter flight via reinforcement learning. In *Experimental robotics IX*, pp. 363–372. Springer, 2006.

Puterman, M. L. *Markov decision processes: discrete stochastic dynamic programming*. John Wiley & Sons, 2014.

Ratliff, N. D., Silver, D., and Bagnell, J. A. Learning to search: Functional gradient techniques for imitation learning. *Autonomous Robots*, 27(1):25–53, 2009.

Ray, A., Achiam, J., and Amodei, D. Benchmarking safe exploration in deep reinforcement learning. *arXiv preprint arXiv:1910.01708*, 7, 2019.

Schulman, J., Wolski, F., Dhariwal, P., Radford, A., and Klimov, O. Proximal policy optimization algorithms. *arXiv preprint arXiv:1707.06347*, 2017.

Scobee, D. R. and Sastry, S. S. Maximum likelihood constraint inference for inverse reinforcement learning. *arXiv preprint arXiv:1909.05477*, 2019.

Silver, D., Bagnell, J. A., and Stentz, A. Learning from demonstration for autonomous navigation in complex unstructured terrain. *The International Journal of Robotics Research*, 29(12):1565–1592, 2010.

Sriperumbudur, B. K., Fukumizu, K., Gretton, A., Schölkopf, B., and Lanckriet, G. R. On integral probability metrics,\phi-divergences and binary classification. *arXiv preprint arXiv:0901.2698*, 2009.

Stooke, A., Achiam, J., and Abbeel, P. Responsive safety in reinforcement learning by pid lagrangian methods. In *International Conference on Machine Learning*, pp. 9133–9143. PMLR, 2020.

Sutton, R. S. and Barto, A. G. *Reinforcement learning: An introduction*. 2018.

Swamy, G., Choudhury, S., Bagnell, J. A., and Wu, Z. S. Of moments and matching: A game-theoretic framework for closing the imitation gap, 2021. URL https://arxiv.org/abs/2103.03236.

Swamy, G., Rajaraman, N., Peng, M., Choudhury, S., Bagnell, J. A., Wu, Z. S., Jiao, J., and Ramchandran, K. Minimax optimal online imitation learning via replay estimation. *arXiv preprint arXiv:2205.15397*, 2022.

Swamy, G., Choudhury, S., Bagnell, J. A., and Wu, Z. S. Inverse reinforcement learning without reinforcement learning. *arXiv preprint arXiv:2303.14623*, 2023.

Todorov, E., Erez, T., and Tassa, Y. Mujoco: A physics engine for model-based control. In *2012 IEEE/RSJ international conference on intelligent robots and systems*, pp. 5026–5033. IEEE, 2012.

Vazquez-Chanlatte, M., Jha, S., Tiwari, A., Ho, M. K., and Seshia, S. Learning task specifications from demonstrations. *Advances in neural information processing systems*, 31, 2018.

Weng, J., Chen, H., Yan, D., You, K., Duburcq, A., Zhang, M., Su, Y., Su, H., and Zhu, J. Tianshou: A highly modularized deep reinforcement learning library. *Journal of Machine Learning Research*, 23(267):1–6, 2022. URL http://jmlr.org/papers/v23/21-1127.html.

Xu, K., Ratner, E., Dragan, A., Levine, S., and Finn, C. Learning a prior over intent via meta-inverse reinforcement learning. In *International conference on machine learning*, pp. 6952–6962. PMLR, 2019.

Xu, M., Liu, Z., Huang, P., Ding, W., Cen, Z., Li, B., and Zhao, D. Trustworthy reinforcement learning against intrinsic vulnerabilities: Robustness, safety, and generalizability. *arXiv preprint arXiv:2209.08025*, 2022.

Yang, T.-Y., Rosca, J., Narasimhan, K., and Ramadge, P. J. Projection-based constrained policy optimization. *arXiv preprint arXiv:2010.03152*, 2020.

Yu, L., Yu, T., Finn, C., and Ermon, S. Meta-inverse reinforcement learning with probabilistic context variables. *Advances in neural information processing systems*, 32, 2019.

Ziebart, B., Dey, A., and Bagnell, J. A. Probabilistic pointing target prediction via inverse optimal control. In *Proceedings of the 2012 ACM international conference on Intelligent User Interfaces*, pp. 1–10, 2012.

Ziebart, B. D., Maas, A. L., Bagnell, J. A., Dey, A. K., et al. Maximum entropy inverse reinforcement learning. In *Aaai*, volume 8, pp. 1433–1438. Chicago, IL, USA, 2008a.

Ziebart, B. D., Maas, A. L., Dey, A. K., and Bagnell, J. A. Navigate like a cabbie: Probabilistic reasoning from observed context-aware behavior. In *Proceedings of the 10th international conference on Ubiquitous computing*, pp. 322–331, 2008b.

Zucker, M., Ratliff, N., Stolle, M., Chestnutt, J., Bagnell, J. A., Atkeson, C. G., and Kuffner, J. Optimization and learning for rough terrain legged locomotion. *The International Journal of Robotics Research*, 30(2):175–191, 2011.

# A. Proofs

## A.1. Proof of Theorem 3.1

*Proof.* Let $\Pi(c)$ denote the set of policies such that

$$J(\pi, c) - J(\pi_E, c) \leq 0. \tag{16}$$

First, note that $\forall c \in \mathcal{F}_c$, $\pi_E \in \Pi(c)$. CRL is therefore trying to maximize reward over a set of policies that contains $\pi_E$. Thus, the reward condition is trivially true. We therefore focus on the safety condition. First, we note that by the definition of regret,

$$\frac{1}{N} \sum_{i}^{N} [(J(\pi_i, \hat{c}) - J(\pi_E, \hat{c})) - (J(\pi_i, c_i) - J(\pi_E, c_i))] = \frac{T}{N} \sum_{i}^{N} \ell_i(\hat{c}) - \ell_i(c_i) \leq \bar{\epsilon} T. \tag{17}$$

This implies that

$$\frac{1}{N} \sum_{i}^{N} [(J(\pi_i, c^*) - J(\pi_E, c^*)) - (J(\pi_i, c_i) - J(\pi_E, c_i))] \leq \bar{\epsilon} T, \tag{18}$$

as $\hat{c}$ is the best-in-hindsight constraint. We then note that $J(\pi_i, c_i) - J(\pi_E, c_i) \leq 0$ by the fact that $\pi_i$ is produced via a CRL procedure, which means we can drop the former term from the above sum, giving us

$$\frac{1}{N} \sum_{i}^{N} (J(\pi_E, c^*) - J(\pi_i, c^*)) \leq \bar{\epsilon} T. \tag{19}$$

Because this equation holds on average, there must be at least one $\pi \in \pi_{1:N}$ for which it holds. Now, we recall that $\pi_i = \mathtt{CRL}(r, c_i)$ to complete the proof. $\qquad\square$

## A.2. Proof of Lemma 3.3

*Proof.* For a single $c$, a standard Hoeffding bound tells us that

$$P(|\frac{1}{K} \sum_{i=0}^{K} V_k(c) - \mathbb{E}[V(c)]| \geq \epsilon) \leq 2 \exp\left(\frac{-2K\epsilon^2}{(2T)^2}\right), \tag{20}$$

where $V_k(c)$ denotes the value of the payoff using data from the $k$th task. We have $|\mathcal{F}_c|$ constraints and want to be within $\epsilon$ of the population mean uniformly. We can apply a union bound to tell us that w.p. at least

$$1 - 2|\mathcal{F}_c| \exp(\frac{-2K\epsilon^2}{(2T)^2}), \tag{21}$$

we will do so. If we want to satisfy this condition with probability at least $1 - \delta$, simple algebra tells us that we must draw

$$K \geq O\left(\log\left(\frac{|\mathcal{F}_c|}{\delta}\right) \frac{(2T)^2}{\epsilon^2}\right) \tag{22}$$

samples. $\qquad\square$

## A.3. Proof of Theorem 3.5

*Proof.* For each $c \in \mathcal{F}_c$, define the set of safe policies as $\Pi(c) = \{\pi \in \Pi | J(\pi, c) \leq 0\}$. This set is non-empty by assumption. Define

$$u(\tau, c) = \mathbb{1}\{\pi_E^\tau \in \Pi(c)\}. \tag{23}$$

We prove each of the desired conditions independently.

**Reward Condition.** Let $\hat{c} \in c_{1:N}$. Recall that we want to prove that

$$\mathbb{E}_{\tau \sim P(\tau)}[J(\pi(r^\tau), r^\tau) - J(\pi_E^\tau, r^\tau)] \geq -2\epsilon T, \tag{24}$$

where $\pi^\tau = \text{CRL}(r^\tau, \hat{c}, \delta = 0)$. Observe that

$$\pi_E^\tau \in \Pi(\hat{c}) \Rightarrow J(\pi^\tau, r^\tau) = \max_{\pi \in \Pi(\hat{c})} J(\pi, r^\tau) \geq J(\pi_E^\tau, r^\tau). \tag{25}$$

Thus, if

$$\mathbb{E}_\tau[u(\tau, \hat{c})] \geq 1 - \epsilon, \tag{26}$$

we have that

$$\mathbb{E}_\tau[J(\pi^\tau, r^\tau) - J(\pi_E^\tau, r^\tau)] \tag{27}$$

$$\leq \mathbb{E}_\tau[(1 - u(\tau, \hat{c}))(J(\pi^\tau, r^\tau) - J(\pi_E^\tau, r^\tau))] \tag{28}$$

$$\leq \mathbb{E}_\tau[(1 - u(\tau, \hat{c}))](\sup_\tau J(\pi^\tau, r^\tau) - J(\pi_E^\tau, r^\tau)) \tag{29}$$

$$\leq \mathbb{E}_\tau[(1 - u(\tau, \hat{c}))]2T \tag{30}$$

$$\leq -2\epsilon T \tag{31}$$

We now prove that with $K$ large enough, we can guarantee Eq. 26 holds true w.h.p. Define

$$\widetilde{\mathcal{F}}_c = \{c \in \mathcal{F}_c | \forall k \in [K], \pi_E^k \in \Pi(\hat{c})\}. \tag{32}$$

We now argue that if

$$K \geq O\left(\log \frac{|\mathcal{F}_c|}{\delta} \frac{1}{\epsilon^2}\right), \tag{33}$$

w.p. $\geq 1 - \delta$, Eq. 26 holds true $\forall c \in \widetilde{\mathcal{F}}_c$. This means that as long as we pick $c_{1:N} \in \widetilde{\mathcal{F}}_c$, our desired condition will be true. Note that this is fewer than the number of samples we assumed in the theorem statement.

For a single constraint, a Hoeffding bound tells us that

$$P(|\frac{1}{K} \sum_k^K \mathbb{1}\{\pi_E^k \in \Pi(c)\} - \mathbb{E}_\tau[u(\tau, c)]| \geq \epsilon) \leq 2\exp\left(-2K\epsilon^2\right). \tag{34}$$

Union bounding across $\mathcal{F}_c \supseteq \widetilde{\mathcal{F}}_c$, we get that the probability that $\exists c \in \widetilde{\mathcal{F}}_c$ s.t. Eq. 26 does *not* hold is upper bounded by

$$1 - 2|\mathcal{F}_c|\exp\left(-2K\epsilon^2\right). \tag{35}$$

To have this quantity be $\geq 1 - \delta$, we need

$$K \geq O\left(\log \frac{|\mathcal{F}_c|}{\delta} \frac{1}{\epsilon^2}\right). \tag{36}$$

**Safety Condition.** We begin by considering the infinite sample setting. We therefore desire to prove that

$$\mathbb{E}_\tau[J(\pi^\tau, c^*) - J(\pi_E^\tau, c^*)] \leq \bar{\epsilon}T. \tag{37}$$

Define the per-round loss of the constraint player as

$$\ell_i(c) = \frac{1}{T}\mathbb{E}_\tau[J(\pi_i^\tau, c) - J(\pi_E^\tau, c)] \in [-1, 1], \tag{38}$$

the best-in-hindsight comparator as $\hat{c} = \text{argmax}_{c \in \mathcal{F}_c} \sum_i^N \ell_i(c)$, instantaneous regret as $\epsilon_i = \ell_i(\hat{c}) - \ell_i(c_i)$, and average regret as $\bar{\epsilon} = \frac{1}{N} \sum_i^N \epsilon_i$. Proceeding similarly to the single-task case,

$$\bar{\epsilon}T = \frac{1}{N} \sum_i^N \mathbb{E}_\tau[J(\pi_i^\tau, \hat{c}) - J(\pi_E^\tau, \hat{c})] - \mathbb{E}_\tau[J(\pi_i^\tau, c_i) - J(\pi_E^\tau, c_i)] \tag{39}$$

$$\geq \frac{1}{N} \sum_i^N \mathbb{E}_\tau[J(\pi_i^\tau, c^*) - J(\pi_E^\tau, c^*)] - \mathbb{E}_\tau[J(\pi_i^\tau, c_i) - J(\pi_E^\tau, c_i)] \tag{40}$$

We now argue that the second term in the above sum must be non-positive. Consider an arbitrary task $\tau$. Then, because $\pi_E^\tau \in \Pi(c_i)$ and CRL is optimizing over $\Pi(c_i)$, this term must be negative. As it is negative per-task, it must be negative in expectation. Thus, we are free to drop the second term in the above expression which tells us that

$$\bar{\epsilon}T \geq \frac{1}{N} \sum_i^N \mathbb{E}_\tau [J(\pi_i^\tau, c^*) - J(\pi_E^\tau, c^*)] \tag{41}$$

Because this equation holds on average, there must be at least one $\pi \in \pi_{1:N}$ for which it holds. Now, we recall that $\pi_i^\tau = \mathrm{CRL}(r^\tau, c_i)$ to complete the infinite-sample proof.

We now consider the error induced by only observing a finite set of tasks. There are two places finite-sample error enters: in estimating the value of $\ell_i(c)$ and in estimating $\widetilde{\mathcal{F}}_c$.

By Lemma 3.3, the maximum error we can induce by estimating $\ell_i$ from finite samples is upper bounded w.h.p by $\epsilon$. Thus, the extra error induced on the average regret is also bounded by $\epsilon$. Observe that our losses are scaled by $\frac{1}{T}$ in comparison to difference of $J$s. Therefore, we need to add an $\epsilon T$ to our bound for the infinite-sample setting.

By our argument in the reward section, $\pi_E^\tau \notin \Pi(c_i)$ w.p. $\leq \epsilon$. When this is true, $J(\pi_i^\tau, c_i) - J(\pi_E^\tau, c_i)$ can be as big as $2T$. Thus, the term we dropped in Eq. 40 ($V(c_i)$) can be as big as $2\epsilon T$ instead of 0. In the worst case, this adds an additional $2\epsilon T$ to our bound.

Combining both of the above, when we transition from the infinite sample setting to the finite sample setting, our bound degrades by $3\epsilon T$. $\qquad\square$

# B. Experimental Details

## B.1. Implementation Details

For the velocity and position constraints, the state of the underlying environment is augmented to include the constraint value (ground-truth constraint for CRL, learned constraint for ICL).

When using CRL as part of ICL, we set the constraint threshold used in the Lagrangian update to be the expert's constraint violation. However, when starting with degenerate constraints, this can prevent policy optimization from learning at all as the expert's violation under the constraint can be arbitrarily low. To circumvent this issue, we set the Lagrangian constraint threshold to use the expert's violation plus a cost limit buffer, which we anneal over the course of training to 0. This ensures that our learned policy satisfies the learned constraint as much as the expert does as desired.

Because ICL requires learning a constraint, we represent our constraints as neural networks, mapping from the state space of our agent to a bounded scalar in the range $[0, 1]$. To update this constraint, we solve the optimization problem using a regression objective. Learner and expert constraint values are labeled with 1 and -1 respectively, and we optimize a regression loss. For both CRL and ICL, we find that using a log-activation on top of the raw value of the constraint is an important detail for stable training.

For the multi-task maze setting, we consider 10 distinct tasks corresponding to unique goal locations, each with 2 starting locations. To reduce the sample-complexity of exploration, we first train four low-level policies, each of which is capable of walking in a particular cardinal direction. Both the expert and learner operate in a discrete action space that corresponds to executing one of these low-level policies. The difference is that the expert gets access to the ground-truth constraint, while the learner gets access to an estimated constraint that is generated by querying our learned constraint network throughout the state space.

We generate expert demonstrations / learner rollouts by combining a waypoint planner with the above low-level controllers. Waypoints are calculated by discretizing the real / estimated maze into a 10 by 10 grid and running Q-value iteration. To generate a trajectory, we follow the sequence of waypoints by using the learned low-level policies to navigate between each pair. Because our low-level policies do not always walk perfectly straight, we enforce the learned constraint once again at rollout time by adding in walls that correspond to the real / estimated structure of the maze.

## B.2. Hyperparameters

All ICL experiments are run with a behavior-cloning initialization, regression objective for constraint updates, and augmenting the environment state with the constraint value.

| Hyperparameter | Value |
| --- | --- |
| PPO Learning Rate | 0.0003 |
| PPO Value Loss Weight | 0.25 |
| PPO Epsilon Clip | 0.2 |
| PPO GAE Lambda | 0.97 |
| PPO Discount Factor | 0.99 |
| PPO Batch Size | 512 |
| PPO Hidden Sizes | [128, 128] |
| P-update Learning Rate | 0.05 |
| I-update Learning Rate | 0.0005 |
| D-update Learning Rate | 0.1 |
| Constraint Batch Size | 4096 |
| Constraint Learning Rate | 0.05 |
| Constraint Update Steps | 250 |
| CRL Epochs | 10 |
| CRL Steps per Epoch | 30000 |
| ICL Expert Demonstrations | 20 |
| ICL Velocity Cost Limit | 20 |
| ICL Position Cost Limit | 100 |
| ICL Anneal Rate | 10 |
| ICL Epochs | 20 |

*Table 1.* Experiment hyperparameters.

