# OpenReview forum: "Learning Shared Safety Constraints from Multi-task Demonstrations"
_ICML.cc/2023/Workshop/ILHF — ILHF Workshop ICML 2023_

### Official Review · Reviewer_7TRD · 2023-06-16
**Learning Shared Safety Constraints from Multi-task Demonstrations**

**Rating:** 7
**Confidence:** 3

**Review:**

The paper is well written and easy to follow, the citations seem adequate and the figures are self-explanatory.

Summary: This paper proposes to learn constraints from expert demonstrations for shared tasks by extending inverse reinforcement learning to the space of constraints. The contributions are 1) formalizing the inverse constraint learning problem, 2) multi-task extension of inverse constraint learning and 3) experiments on continuous control tasks. There are 3 algorithms introduced - constrained RL, inverse constraint learning, and multi-task inverse constraint learning. ICL is framed as a zero-sum game between two agents (policy learner and the constraint learner). The policy learner maximizes reward while satisfying potential constraints, whereas the constraint player learns constraints that maximally penalizes the learner relative to the expert.  Given the ill-posed problem, this paper further proposes using multi-task demonstrations to learn safety constraints.

The paper addresses an important problem in inverse reinforcement learning. All the theoretical contributions look reasonable and the simulation experiments seem adequate as initial set of experiments to verify the proposed approach. Fig. 4 is hard to understand, and needs annotations in the image to understand the scene. Simulation experiments look promising and I encourage the authors to consider real-robot experiments in future submissions to strengthen the contribution further.

---

### Decision · Program_Chairs · 2023-06-20

Accept